# Hypomagnesemia and the Metabolic Syndrome among Apparently Healthy Kuwaiti Adults: A Cross-Sectional Study

**DOI:** 10.3390/nu14245257

**Published:** 2022-12-09

**Authors:** Dalal Alkazemi, Noora Alsouri, Tasleem Zafar, Stan Kubow

**Affiliations:** 1Department of Food Science and Nutrition, College of Life Sciences, Kuwait University, Shadadiya 12037, Kuwait; 2School of Human Nutrition, McGill University, Montréal, QC H9X 3V9, Canada

**Keywords:** hypomagnesemia, metabolic syndrome, prevalence, cardiometabolic

## Abstract

Magnesium plays a key role in metabolic disorder development, and hypomagnesemia may be implicated in the pathogenesis of metabolic syndrome (MetS) and its components. In this cross-sectional study, we investigated the associations between hypomagnesemia, MetS, and MetS components among 231 adults (193 women and 38 men) living in Kuwait who were apparently healthy without chronic diseases. We used the International Diabetes Federation (IDF) and the United States National Cholesterol Education Program Adult Treatment Panel III (ATP III) criteria to define participants with MetS. The Ministry of Health cutoff for hypomagnesemia (<0.74 mmol/L) was employed. IDF- and ATP III-defined MetS prevalence was 22.1% and 15.2%, respectively. Hypomagnesemia occurred in 33.3% of all participants and 53.2% of participants with MetS (*p* < 0.001). Magnesemia correlated negatively with body mass index, waist circumference, systolic blood pressure [SBP], diastolic blood pressure (DBP), fasting blood glucose (FBG), low-density lipoprotein cholesterol level, and triglyceride level; magnesemia correlated positively with high-density lipoprotein cholesterol (HDL-C) levels (*p* < 0.001). Multivariate logistic regression, adjusting for BMI, age, and sex, showed that hypomagnesemia was associated with a 12- and 5-fold greater odds of getting IDF-defined (adjusted odds ratio [aOR] 11.70; 95% confidence interval [CI] 4.87–28.14) and ATP-defined (aOR 5.44; 95% CI 2.10–14.10) MetS, respectively, in the study population. Hypomagnesemia was significantly associated with a 3.62, 9.29, 7.01, 2.88, 3.64, and 3.27 higher odds of an increased waist circumference (95% CI 1.48–8.85), elevated serum triglyceride level (95% CI 3.97–21.73), elevated FBG (95% CI 3.25–15.11), elevated SBP (95% CI 1.16–7.11), elevated DBP (95% CI: 1.22–10.89), and lowered HDL-C level (95% CI 1.69–6.32), respectively. Hypomagnesemia could be a consequence of the pathophysiology of MetS and its individual components among adults in Kuwait.

## 1. Introduction

Magnesium, an essential mineral and one of the most abundant intracellular cations, is a cofactor for enzymes involved in many cellular processes, including carbohydrate and energy metabolism [1]. Evidence suggests that magnesium deficiency (hypomagnesemia) may play a role in the pathophysiology of a cluster of metabolic disorders, including obesity, insulin resistance, type 2 diabetes mellitus (T2DM), metabolic syndrome (MetS), and cardiovascular disease (CVD) [2,3]. Magnesium depletion is commonly reported in patients with diabetes, and the onset and progression of diabetic complications are associated with magnesium homeostasis impairment [4]. Hypomagnesemia has been associated with the components of MetS, including hyperglycemia, hypertension, hypertriglyceridemia, and insulin resistance [5]. Magnesium supplementation reportedly demonstrates good efficacy on the components of MetS, including insulin sensitivity, fasting blood glucose (FBG), triglyceride (TG) levels, high-density lipoprotein cholesterol (HDL-C) levels, and high blood pressure (BP) [6]. Chronic low-grade inflammation is a shared critical pathophysiological component in the abovementioned metabolic disorders; moreover, magnesium depletion can, directly and indirectly, induce inflammation by modifying the intestinal microbiota [7]. In addition, magnesium deficiency promotes inflammation by dysregulating the immunological function through priming phagocytes, enhancing granulocyte oxidative burst, activating endothelial cells, and increasing the levels of cytokine production [5]. The protective actions of magnesium include anti-inflammation, glucose and insulin metabolism improvements, endothelium-dependent vasodilation enhancement, and lipid profile normalization [8]. Serum magnesium level is inversely related to the serum level of C-reactive protein, a marker of systemic inflammation and a predictor of future cardiovascular events in patients with MetS [9]. In clinical practice, serum magnesium level is the most used marker of magnesium deficiency, and there is a consensus that hypomagnesemia is a definite sign of magnesium deficiency [10]. The current global reference range for serum magnesium level is 0.75–0.95 mmol/L. Epidemiological data have shown that patients with serum magnesium levels below 0.75–0.85 mmol/L demonstrate an increased risk of CVD and T2DM, as well as mortality from these diseases [10].

In Kuwait, serum magnesium level is typically a part of clinical investigations; however, to our knowledge, no study has hitherto described serum magnesium levels in a Kuwaiti sample in relation to MetS, which places an individual at a high risk of developing CVD and T2DM [11]. Previous investigations of MetS in Kuwait, using either criteria of the Adult Treatment Panel III (ATP III) or International Diabetes Federation (IDF), showed a high prevalence of MetS. Al Zenki et al. [12] found that 37.7% and 40.1% of Kuwaiti women aged ≥20 years had ATP III- or IDF-defined MetS, respectively. Sorkhou et al. [13] reported an overall ATP III-defined MetS prevalence of 34% (28.7% in men and 39.7% in women) among Kuwaiti hypertensive patients aged ≥40 years. A lower prevalence of MetS according to ATP III at 18% was noted among apparently healthy adults with no prior chronic disease, diabetes, or heart disease [14]. Among adolescents aged 10–19 years, the prevalences of IDF- versus ATP III-defined MetS were 14.8% versus 19.5% in boys and 14.8% versus 9.1% in girls, respectively [15,16]. The proportion of individuals with MetS is projected to continuously increase in parallel with the prevalence of obesity and T2DM. In this study, we aimed to assess the prevalence of hypomagnesemia in a sample of apparently healthy adults living in Kuwait and to assess the association between hypomagnesemia and MetS and its cardiometabolic components before the development of overt chronic diseases, including diabetes and heart disease. The early establishment of these associations underscores the importance of magnesium status as an early biomarker of metabolic dysregulations. The rationale for these selection criteria was to enable the investigation of magnesium status before the confounding interference of the disease-based physiological disruptions and medication use on magnesium homeostasis, which is well described in the literature. This approach limits the impact of the above confounders that can obscure the early role of magnesium in MetS development. To accomplish the overall research goal, our specific study objectives were (1) to examine the sample prevalence of IDF- and ATP III-defined MetS among a sample of apparently healthy adults free of chronic diseases and not on any prescription medications; (2) to assess the serum magnesium levels in those with and without MetS; and (3) to determine the associations between serum magnesium levels and cardiometabolic factors.

## 2. Materials and Methods

### 2.1. Study Design, Recruitment, and Participant Selection

This was a cross-sectional study, with a convenient sample of participants obtained from two major hospitals (Al-Addan Hospital and Mubarak Al-Kabeer Hospital) representing residents of two (Al-Ahmadi and Hawalli) out of six governorates in Kuwait. Participant recruitment and data collection were performed at the outpatient clinics and medical laboratories between May 2018 and January 2019. The sample size was a priori calculated based on the previously reported prevalence of MetS among apparently healthy adults and ranged from 198–265 participants [14,17,18]. We targeted individuals who either came for regular checkups or were referred by their general physicians for medical investigations without diagnosed diabetes or heart conditions. The target group was invited to participate in the study on a voluntary basis. Individuals who fulfilled the inclusion criteria were asked to complete a self-administered questionnaire to collect socio-demographic (sex, age, nationality, level of education, total income, and occupation) and clinical (the presence of chronic disease, physical activity, family history of diseases, smoking, and the use of any medication or supplement) data. Thereafter, all the participants underwent anthropometric and biochemical measurements. Participants were informed that they could withdraw from the study at any time without having to justify their withdrawal, and this would not affect their medical care. All participants provided written informed consent before enrollment in the study. Ethical approval was obtained from the Kuwait University research sector and the Ministry of Health (MOH). We included residents in Kuwait (both Kuwaiti and non-Kuwaiti nationals), aged 18–65 years, with no chronic disease history, and without the use of prescription medications. We excluded individuals with diabetes, renal failure, or pregnancy; nursing women; those taking prescription medications for hypertension and dyslipidemia; those taking any type of fiber or magnesium supplements; and those who did not fast overnight (10–12 h).

### 2.2. MetS Diagnostic Criteria

The IDF criteria and the United States National Cholesterol Education Program ATP III criteria were used to define patients with MetS [19]. According to the IDF criteria, MetS was diagnosed based on waist circumference ≥ 94 cm for men and ≥80 cm for women plus two of the following criteria: elevated TG level ≥ 150 mg/dL (1.7 mmol/L), reduced HDL-C level < 40 mg/dL (1.03 mmol/L) in men and <50 mg/dL (1.29 mmol/L) in women, high BP (systolic BP [SBP] ≥ 130 or diastolic BP [DBP] ≥ 85 mm Hg), and increased FBG ≥ 100 mg/dL (5.6 mmol/L). According to the modified ATP III, MetS was diagnosed when individuals met at least three of the following criteria: abdominal obesity (waist circumference > 102 cm [40 in] in men and >88 cm [35 in] in women, elevated TG level ≥ 150 mg/dL (1.7 mmol/L), reduced HDL-C level < 40 mg/dL (1.03 mmol/L) in men and <50 mg/dL (1.29 mmol/L) in women, high BP (SBP ≥ 130 mmHg and DBP ≥ 85 mmHg (or on treatment for hypertension), and increased FBG ≥ 100 mg/dL (5.6 mmol/L) (or on treatment for diabetes).

### 2.3. Anthropometric and Blood Pressure Measurements

Weight was measured using digital scales, with the participant wearing light clothes and without shoes. Height was measured using a calibrated stadiometer while the participant stood with the shoulders in a normal position. Body mass index (BMI) was calculated as weight (in kg) divided by height squared (in m^2^). Finally, waist circumference was measured (to the nearest 0.1 cm) at a level midway between the inferior margin of the ribs and the superior border of the iliac crest, using a non-stretchable measuring tape, with the participant wearing light clothes. A nurse took three BP readings using a standard-gauge mercury sphygmomanometer placed on the right arm, with the participant in a seated position and after a complete rest. The mean of the three readings was taken as the BP value.

### 2.4. Biochemical Measurements

After an overnight fasting period of 10–12 h, the participants came to the hospital the next morning for blood sample collection. A trained phlebotomist performed venipuncture for each participant. The participants were seated for 5 min before blood collection from the left arm. One blood sample was collected for each participant using a 3.5-mL yellow-top plain tube. The blood specimens were immediately sent to the laboratory for processing, starting with centrifugation using either Rotofix 32 A or Thermo centrifuge machines. After centrifugation, the specimens were analyzed using automated Roche Cobas 8000 c702, which is a specialized photometric measuring unit for analyzing FBG, TG, HDL-C, low-density lipoprotein cholesterol (LDL-C), total cholesterol, and serum magnesium levels. The reagents used for analysis were supplied by the same manufacturer [“eLabDoc-Roche Dialog”, Indianapolis, IN, USA, 2018]. In this study, we used the normal reference ranges recommended by the MOH. Based on the MOH population-specific reference ranges, hypomagnesemia was defined as a serum magnesium level < 0.74 mmol/L.

### 2.5. Statistical Analysis

Statistical analysis was performed using SPSS version 25.0 (IBM, Chicago, IL, USA). The Kolmogorov–Smirnov test was used to assess the normality of continuous variables. The Mann–Whitney U-test for non-normal variables was used for comparisons between participants with and without MetS, defined based on IDF criteria, as for all the analyses. The relationship between continuous variables was examined using Spearman linear correlation analysis. A binary logistic regression analysis was conducted to examine whether hypomagnesemia (independent variable) was related to MetS occurrence. Finally, logistic regression models, including variables for age, nationality, education level, employment status, total annual income, menstrual cycle (for women), BMI, physical activity, and serum magnesium level were used to estimate prevalence odds ratios (ORs) for factors that were associated with MetS occurrence. The covariates for MetS were determined using univariate analyses, and covariates that demonstrated statistical significance were included in the final multivariate logistic model. A *p*-value < 0.05 was considered statistically significant.

## 3. Results

### 3.1. Participant Characteristics

A total of 247 agreed to participate in the study. We excluded 13, 2, and 1 participants for uncompleted questionnaires or inappropriate fasting duration, existing diabetes, and pregnancy, respectively; hence, 231 participants (193 women and 38 men) were included in the final analysis. Table 1 shows the overall and sex-related characteristics of the study participants. Serum magnesium level was in the low range in the overall sample population.

### 3.2. Prevalence of MetS

The prevalence of IDF- and ATP III-defined MetS in the overall sample population was 22.1% (23.3% in women and 15.8% in men, *p* = 0.394) and 15.2% (16.6% in women and 7.9% in men, *p* = 0.220), respectively. The prevalence of MetS increased with age as follows: 11.7%, 47.1%, and 92.3% among participants aged 18–29, 30–49, and 50–65 years, respectively (*p* < 0.001) (Table 2). The age-adjusted prevalence of MetS according to IDF for the overall sample increased from 22.1% to 27.4% (from 23.3% to 28.3% in females and from 15.8% to 23% in males). Similarly, when using ATP III, the overall MetS prevalence increased from 15.2% to 19.1% (from 16.6% to 20.6% in females and from 7.9% to 11.9% in males). In addition, the prevalence of MetS increased with weight status (3.6%, 37.8%, 51.6% among participants with normal weight, overweight, and obesity, respectively [*p* < 0.001]). The prevalence of MetS was higher in women who reported having irregular menstrual periods than in those with regular periods (54.2% vs. 13.1%, *p* < 0.001). Moreover, the prevalence of MetS was higher in menopausal women than in those with normal menstruation (63.6% vs. 20.9%, *p* = 0.001). The prevalence of MetS did not differ by annual income (22.7% [<5000 KD] vs. 26.5% [≥5000 KD], *p* = 0.630), employment status (22.1% [employed] vs. 22.1% [unemployed]), educational level (21.1% [high school or lower] vs. 22.4% [college degree or higher]), marital status (26.6% [married] vs. 17.9% [single] or 20.1% [divorced], *p* = 0.289), or nationality (22.9% [Kuwaiti] vs. 16.7% [non-Kuwaiti], *p* = 0.444). The prevalence of hypomagnesemia was significantly higher in participants with MetS than in those without MetS (53.2 vs. 46.8%, *p* < 0.001) (Table 2).

### 3.3. Cardiometabolic Risk Factors by Sex

More women than men had an obesogenic waist based on both IDF (60.6% vs. 26.3%, *p* < 0.001) and ATP III (24.9% vs. 7.9%, *p* = 0.021) criteria (Table 3). Furthermore, more women had low HDL-C levels than men (37.3% vs. 13.2%, *p* = 0.004). However, more men had a higher SBP than women (28.9% vs. 13.5%, *p* = 0.017). There was no statistically significant difference between men and women in terms of the other cardiometabolic risk factors (Table 3). Hypomagnesemia was present in 33.3% of the overall sample population, with no significant difference in hypomagnesemia prevalence between women (32.1%) and men (39.5%).

### 3.4. Hypomagnesemia and MetS

The prevalence of MetS was higher among participants with hypomagnesemia than in those with normal serum magnesium levels (53.2% vs. 6.5%, *p* < 0.001) (Table 2). Participants with MetS had significantly lower median serum magnesium levels compared to those without MetS (median [interquartile range]: 0.70 [0.63–0.74] vs. 0.81 [0.76–0.90], *p* < 0.001) (Table 4). In addition, participants with MetS were older and weighed more than those without MetS (Table 4). All cardiometabolic risk factor parameters were significantly higher in participants with MetS than in those without MetS (Table 4).

### 3.5. Correlations of Serum Magnesium Level with Cardiometabolic Risk Factors

Serum magnesium level showed a significant negative association with all the cardiometabolic risk factors (*p* < 0.001) (Table 5), except for HDL-C level, which showed a positive association (r = 0.368, *p* < 0.001). The strongest associations were demonstrated between serum magnesium level and BMI (r = −0.423, *p* < 0.001), waist circumference (r = −0.412, *p* < 0.001), FBG (r = −0.517, *p* < 0.001), and TG level (r = −0.586, *p* < 0.001).

### 3.6. Correlations of Hypomagnesemia with MetS and Its Components

Table 6 shows the associations between hypomagnesemia and having IDF- and ATP III-defined MetS, using both unadjusted and adjusted models. In the overall population, participants with hypomagnesemia showed a 12-fold increase in the odds of having IDF-defined MetS (adjusted OR [aOR] 11.70; 95% confidence interval [CI] 4.87–28.14). In men with hypomagnesemia, only the unadjusted model showed an 11-fold increase in the odds of having MetS [OR 11.00; 95% CI 1.133–106.84]. Women with hypomagnesemia demonstrated a 13-fold increase in the odds of having MetS [aOR 13.36; 95% CI 5.27–33.83]. Hypomagnesemia was associated with a five-fold increase in the odds of having ATP III-defined MetS in the overall sample population [aOR 5.44; 95% CI 2.10–14.10] and in women [aOR 5.27; 95% CI 1.99–13.99]. Both unadjusted and adjusted models showed no association between hypomagnesemia and MetS occurrence in men. Table 7 presents the associations between hypomagnesemia and MetS components, using both unadjusted and adjusted models. Hypomagnesemia was significantly associated with a 3.62, 9.29, 7.01, 2.88, 3.64, and 3.27 higher odds of having a higher waist circumference (95% CI 1.48–8.85), elevated serum TG level (95% CI 3.97–21.73), elevated FBG (95% CI 3.251–15.105), elevated SBP (95% CI 1.16–7.11), elevated DBP (95% CI: 1.22–10.89), and lowered HDL-C level [95% CI: 1.698–6.315], respectively.

## 4. Discussion

This study clearly showed that hypomagnesemia is significantly associated with the occurrence of MetS and its components in apparently healthy adults in Kuwait. Participants with hypomagnesemia were eleven and five times more likely to have IDF- and ATP III-defined MetS, respectively. The prevalence of hypomagnesemia was significantly higher among participants with MetS than among those without MetS. Moreover, patients with MetS had serum levels of magnesium below the internationally accepted reference range of 0.75–0.95 mmol/L [20]. Clinically, individuals with magnesium levels < 0.8 mmol/L are likely to have severe magnesium deficiency in tissues and bones [21]. This is because the human serum magnesium level normally represents only 0.8%, of which 0.3% and 0.5% are in the serum and erythrocytes, respectively. Furthermore, magnesium is kept under tight regulations from body pools, mainly the bones and teeth (53%), as well as the intercellular space, muscles, and soft tissues (46%) [22]. Magnesium levels in the body can be influenced by factors that affect the total amount of magnesium absorbed and excreted through the kidneys, such as dietary intake or supplementation and serum albumin level [21]. In our overall sample population, 33.3% of participants had subclinical magnesium deficiency based on serum magnesium level, which corroborates with estimates obtained from population-based data (10 to 30%) [21]. In addition, other comorbidities such as inflammatory bowel disease, poorly controlled diabetes, or renal disease exacerbate magnesium deficiency [23]. This was not a point of concern in our study, as participants with overt disease were excluded.

In this study, we found that the prevalence of IDF- and ATP III-defined MetS was 22% and 15% in the overall sample population, respectively. The discrepancy in MetS prevalence based on the two definitions used is consistent with that observed in other investigations, with IDF-defined MetS showing a higher prevalence than ATP III-defined MetS. This may be due to the lower cutoff point values for waist circumference used in the IDF. For example, the waist circumference in females is ≥80 cm and >88 cm based on IDF and ATP III criteria, respectively. Until more specific data are available from this region, the European WC cutoff points are used for Middle Eastern and Arab countries [19]. Studies in Iran, Oman, Iraq, Tunisia, and Egypt proposed various population-specific cutoff point values reflective of differences in fat distribution patterns according to sex and between ethnicities, as suggested by the IDF [24]. Such an endeavor may improve the estimation of abdominal obesity and MetS prevalence among the Kuwaiti population.

The MetS prevalence obtained in this study was comparable to that obtained in previous studies conducted on similar participants in Kuwait or neighboring Gulf Cooperation Council (GCC) countries. For example, in Kuwait, Badr et al. reported an ATP III-defined MetS prevalence of 18% among 434 healthy adults aged 20–44 years recruited from Qurtuba and Abdullah Al-Salem primary health centers/polyclinics [14]. The authors defined “healthy participants” as those without chronic diseases such as diabetes, hypertension, heart problems, and dyslipidemia, as well as those who were not pregnant. In the Kingdom of Saudi Arabia, Alzahrani et al. [25] reported an overall age-adjusted ATP III-defined MetS prevalence of 21% (22.8% in men vs. 13.8% in women), in a sample of 600 healthy Saudi adults aged 20–50 years without chronic diseases or pregnancy. In Oman, Al-Lawati et al. demonstrated that the crude and age-adjusted prevalence of ATP III-defined MetS among Omani adults aged ≥ 20 years was 17% and 21%, respectively; the age-adjusted prevalence in men was 19.5% and 23% in women [26]. These similarities in MetS prevalence among GCC countries may be due to the similar lifestyles and dietary habits among the populations [27]. Al-Zenki et al. and Malik et al. reported a higher MetS prevalence in the Kuwaiti population, probably due to differences in sampling criteria, as they included participants from the general population and not only healthy adults [12,28]. Individuals with existing health conditions (such as diabetes, hypertension, and other diseases) are predisposed to developing MetS.

We found a higher MetS prevalence among older participants; furthermore, overweight and obese participants showed a higher MetS prevalence than participants who were a normal weight. These findings are similar to those reported in Kuwait by Al Zenki et al. [12], who found that individuals aged 40–59 and ≥60 years were more than 2.5 and 4.9 times likely to develop MetS than those aged 20–39 years, respectively. Moreover, Kuwaiti patients with obesity and who were overweight were 19.7 and 4.8 times more likely to develop MetS compared with individuals with normal BMIs, respectively [12]. The increased prevalence of obesity in Kuwait and globally is paralleled with an increased number of individuals with MetS and CVD [29]. Obesity induces all major metabolic disorders, especially diabetes, hypertension, dyslipidemia, and CVD [30].

We also showed that the IDF- and ATP III-defined prevalence of MetS was slightly higher in women (23.3% and 16.6%) than in men (15.8% and 7.9%), respectively, and women demonstrated stronger associations between MetS occurrence and hypomagnesemia, even after adjusting for age and BMI. These sex-related differences may be linked to the higher prevalence of a obesogenic waist in women due to more excessive visceral fat deposition; the seemingly higher BMI in men may reflect variations in body composition. In agreement with our findings, Oguoma et al. [30], in a population-based cross-sectional survey of adults living in Kuwait, showed that the odds of being overweight were 26% greater for men than for women; however, women had a 54% and seven-fold greater odds of developing obesity/central obesity than men, demonstrating greater adiposity in women. Abdominal obesity has been linked to direct adipokine alterations, which play an important role in the metabolic homeostasis of healthy individuals [31]; this indirectly promotes insulin resistance and MetS development by increasing the release of free fatty acids to the liver and inflammatory mediator secretion [31]. Changes in magnesium metabolism in patients with obesity trigger a reduction in serum and erythrocyte magnesium levels [8]. We found significant negative associations of serum magnesium level with waist circumference and BMI. Similarly, a population-based cross-sectional study reported significant negative correlations of body weight and waist circumference with total serum magnesium levels in 130 healthy adults [32].

Furthermore, the effects of pregnancy and menopause may explain the increased susceptibility of women to MetS and its components. During pregnancy, the body weight might increase by more than 20% [33], gestational diabetes may increase the risk of T2DM and MetS [34], and preeclampsia may predispose the woman to hypertension and CVD later in life [35]. Additionally, elevated LDL-C, HDL-C, total cholesterol, and TG levels may occur, especially during the second and third trimesters of pregnancy [36]. Previous studies have demonstrated that the menopause may promote changes in body fat distribution, which increase central adiposity [37]; in addition, the menopause is significantly associated with high levels of total cholesterol, LDL, VLDL, and TG, as well as low HDL levels compared with the premenopausal status [38]. These pregnancy- and menopause-related changes contribute to increasing cardiovascular death rates among women under 55 years, and thus early screening for CV risk factors and lifestyle modification should be implemented [35]. Overeating and a sedentary lifestyle worsen with age and mediate MetS by increasing the accumulation of central adiposity and ectopic fat infiltration in the skeletal muscle and the liver, driving the development of insulin resistance with ectopic fat accumulation, magnesium metabolism alterations, systemic and hypothalamic inflammation, shortening of telomere length, dysregulating epigenetic mechanisms, and disturbing the circadian rhythm [39]. Several population-based studies showed that older adults (aged > 65 years) had a higher prevalence of MetS than younger age groups, and that the MetS factors were equally prevalent in men and women [39,40]. It was suggested that the diminished sex differences in the metabolic risk profile might be due to the diminished sex differences in total and visceral adiposity with age and the cardiometabolic effects of the menopause. However, there appeared important sex and age differences in the way the different metabolic syndrome combinations relate to CVD morbidity and mortality risk [40]. For example, although low HDL-C was one of the most common metabolic syndrome components in younger women and among the least common in older women, it was one of the more robust correlates of mortality risk in both age strata [38]. Moreover, the association between MetS and mortality risk did not appear to be related to the number of MetS components one displays [40,41]. More research with comprehensive data is needed to determine the specific combinations of metabolic syndrome components predictive of CVD morbidity and mortality risk in Kuwait.

Our data clearly show an association between serum magnesium level and all cardiometabolic risk factors, as expected; moreover, hypomagnesemia increased the odds of having these risk factors, independently from age, sex, and BMI. Serum magnesium level was significantly associated with components such as a low HDL-C level, high waist circumference, high TG level, high FBG, and high SBP. Our study could not clarify if hypomagnesemia preceded or resulted from MetS development. Hypomagnesemia can promote cell dysfunction and diminish vascular tone and resistance by increasing nitric oxide release, which antagonizes the effect of vasoconstrictor molecules such as calcium, bradykinin, angiotensin II, or serotonin, thereby increasing the BP [42]. Extracellular magnesium level reduction, intracellular calcium level increase, and phagocytic cell priming trigger inflammatory cytokine release [43]. Subclinical magnesium deficiency can promote chronic inflammatory reactions and abnormalities in cell signaling, thereby contributing to the release of inflammatory molecules such as neuropeptides, cytokines, prostaglandins, and leukotrienes. This leads to impaired insulin secretion, insulin resistance, and hyperlipidemia, which promote MetS development [8]. Chronic low magnesium intake has been shown to cause serum and intracellular magnesium deficiency. Hypomagnesemia induced by poor dietary intake leads to chronic low-grade inflammation and the release of inflammatory cytokines and free radicals, which are the underlying mechanisms for insulin resistance, hypertension, and dyslipidemia [44]. As reviewed by Piuri et al. [45], several investigations showed a positive correlation between low dietary magnesium intake and MetS risk, independently from other risk factors such as age, sex, BMI, race, educational attainment, marital status, smoking, alcohol intake, exercise, energy intake, percentage of calories from saturated fat, and the use of an antihypertensive or lipid medication.

To our knowledge, this is the first study investigating the association between serum magnesium level and MetS in Kuwait and the Gulf region. We used both IDF and ATP III criteria to diagnose MetS, which further strengthened our study and increased the chances of comparison with other studies. An adequate sample size was obtained to estimate the sample prevalence of MetS with 5% precision and to provide powered analyses.

Our study has a few limitations. The cross-sectional study design cannot infer causality. In addition, the study included a small sample size that covered only the Al-Ahmadi and Hawalli governorates due to difficulties with recruitment and subject retention. Thus, the sample is not representative of the entire Kuwait population. Moreover, we did not assess the effect of dietary magnesium and related nutrient (fiber and calcium) intake, as well as physical activity. Hypokalemia and hypocalcemia are seen with poor magnesium status, and so the lack of the above measures could be a limitation. Future work could more comprehensively assess magnesium status, the oral magnesium load test, food records of Mg intake, and urinary magnesium. Finally, there was a possibility for selection bias since study participation was completely voluntary; the study participants might have been more health conscious.

## 5. Conclusions

In women, a high waist circumference and low HDL-C levels were the most frequent components of MetS; therefore, public health measures should be implemented to prevent and control obesity among adult women. Caloric control, improving diet quality, and increasing caloric expenditure should be encouraged through patient-centered programs. Further studies should be conducted to determine appropriate and effective interventions for improving magnesium status, the association of other mineral intakes (such as calcium and potassium) and MetS occurrence, and the effect of magnesium supplementation in reducing MetS occurrence in the Kuwaiti population. In addition, sex- and age-specific waist circumference cutoff points are required for the Kuwaiti population to improve the estimation of MetS prevalence. Finally, prospective randomized trial studies, with representative samples from all six governorates in Kuwait, are needed to investigate the effect of dietary magnesium and fiber intake on the prevalence of MetS and its components. Early MetS identification and interventions will help in preventing further complications, including diabetes and CVDs, and thus lowering mortality rate [46].

## Figures and Tables

**Table 1 nutrients-14-05257-t001:** Participant characteristics.

Characteristics		Overall	Women	Men
		*n* = 231	*n* = 193 (83.5%)	*n* = 38 (16.5%)
Age (years)	Mean ± SD	29.76 ± 9.45	29.41 ± 9.48	31.53 ± 9.19
		*n* (%)		
Nationality	Kuwaiti	201 (87)	168 (87)	33 (86.8)
	Non-Kuwaitis	30 (13)	25 (13)	5 (13.2)
Marital status	Single	112 (48.5)	98 (50.8)	14 (36.8)
	Married	109 (47.2)	85 (44)	24 (63.2)
	Divorced	10 (4.3)	10 (5.2)	--
Educational level	High school or lower	57 (24.7)	44 (22.8)	13 (34.2)
	College degree or higher	174 (75.3)	149 (77.2)	25 (65.8)
Employment	Yes	136 (58.9)	106 (54.9)	30 (78.9)
	No	95 (41.1)	87 (45.1)	8 (21.1)
Annual family income	<5000 KD	44 (19)	40 (20.7)	4 (10.5)
	5000 KD or more	98 (42.4)	70 (36.3)	28 (73.7)
	Did not disclose	89 (38.5)	83 (43)	6 (15.8)
Smoking	Yes, current	7 (3)	--	7 (18.4)
	No, never	222 (96.1)	193 (100)	29 (76.3)
	No, but used to	2 (0.9)	--	2 (5.3)
Regular periods	Yes	--	152 (78.8)	--
	No	--	41 (21.2)	--
Menopause	Yes	--	11 (5.7)	--
	No	--	182 (94.3)	--
Hormones/BC	Yes	--	15 (7.8)	--
	No	--	178 (92.2)	--
Family History				
Diabetes	Yes	160 (69.3)	130 (67.4)	30 (78.9)
HTN	Yes	132 (57.1)	119 (61.7)	13 (34.2)
Heart disease	Yes	39 (16.9)	36 (18.7)	3 (7.9)
Weight status	Underweight	8 (3.5)	7 (3.6)	1 (2.6)
	Normal weight	110 (47.6)	95 (49.2)	15 (39.5)
	Overweight	82 (35.5)	67 (34.7)	15 (39.5)
	With obesity	31 (13.4)	24 (12.4)	7 (18.4)

Abbreviation: BC, birth control pills, KD, Kuwaiti dinar; HTN, hypertension.

**Table 2 nutrients-14-05257-t002:** Sample prevalence of MetS based on IDF and ATP III criteria.

Characteristics		IDF Criteria			ATP III Criteria		
		With MetS	Without MetS	Sig *	With MetS	Without MetS	Sig *
Overall		51 (22.1)	180 (77.9)		35 (15.2)	196 (84.8)	
Age-adjusted		(27.4)	(72.6)		(19.1)	(80.9)	
Sex	Female	45 (23.3)	148 (76.7)	0.307	32 (16.6)	161 (83.4)	0.172
	Male	6 (15.8)	32 (84.2)		3 (7.9)	35 (92.1)	
Age-adjusted	Female	(28.3)	(71.75)	<0.001	(20.6)	(79.4)	<0.0001
	Male	(23.0)	(77.0)		(11.9)	(88.1)	
Age group	18–29 years	15 (11.7)	113 (88.3)	<0.001			
	30–49 years	24 (26.7)	66 (73.3)				
	50–65 years	12 (92.3)	1 (7.7)				
Weight status	Underweight	--	8 (100)	<0.001			
	Normal	4 (3.6)	106 (96.4)				
	Overweight	31 (37.8)	51 (62.2)				
	Obesity	16 (51.6)	15 (48.8)				
Irregular menstrual period	Yes	26 (54.2)	22 (45.8)	<0.001			
	No	19 (13.1)	126 (86.9)				
Menopause	Yes	7 (63.6)	4 (36.4)	<0.001			
	No	38 (20.9)	144 (79.1)				
Family annual income	<5000 KD	10 (22.7)	34 (77.3)	0.630			
	≥5000 KD	26 (26.5)	72 (73.5)				
Educational level	High school or lower	12 (21.1)	45 (78.9)	0.830			
	College degree or higher	39 (22.4)	135 (77.6)				
Marital status	Single	20 (17.9)	92 (82.1)	0.289			
	Married	29 (26.6)	80 (73.4)				
	Divorced	2 (20.0)	8 (80.0)				
Nationality	Kuwaiti	46 (22.9)	155 (77.1)	0.444			
	Non-Kuwaiti	5 (16.7)	25 (83.3)				
Serum magnesium level	<0.74 mmol/L	41 (53.2)	36 (46.8)	<0.001			
	>0.74 mmol/L	10 (6.5)	144 (93.5)				

* Pearson Chi-Squared Test, Asymptotic Significance, 2-sided. Abbreviations: IDF, International Diabetes Federation; ATP III, Adult Treatment Panel III.

**Table 3 nutrients-14-05257-t003:** Distribution of cardiometabolic risk factors in the overall sample population in terms of sex.

Cardiometabolic Risk Factors	Normal Reference Ranges **	Cutoff Point * for At-Risk Category	*n* (%)		
			Overall, *n* = 231	Female, *n* = 193	Male, *n* = 38
WC (cm), IDF		Obesogenic: Men ≥ 94, Women ≥ 80	127 (55)	117 (60.6)	10 (26.3)
		Normal	104 (45)	76 (39.4)	28 (73.7)
WC (cm), ATP III		Obesogenic: Men > 102, Women > 88	51 (22.1)	48 (24.9)	3 (7.9)
		Normal	180 (77.9)	145 (75.1)	35 (92.1)
TG level (mmol/L)	0.56–1.31	Elevated ≥ 1.69	44 (19)	38 (19.7)	6 (15.8)
		Normal	187 (81)	155 (80.3)	32 (84.2)
FBG (mmol/L)	3.9–6.1	Elevated ≥ 5.6	51 (22.1)	42 (21.8)	9 (23.7)
		Normal	180 (77.9)	151 (78.2)	29 (76.3)
SBP (mmHg)	90–120	Elevated ≥ 130	37 (16)	26 (13.5)	11 (28.9)
		Normal	194 (84)	167 (86.5)	27 (71.1)
DBP (mmHg)	60–80	Elevated ≥ 85	24 (10.4)	18 (9.3)	6 (15.8)
		Normal	207 (89.6)	175 (90.7)	32 (84.2)
LDL-C level (mmol/L)	2.1–4.10	Elevated ≥ 4.12	13 (5.6)	12 (6.2)	1 (2.6)
		Normal	218 (94.4)	181 (93.8)	37 (97.4)
HDL-C level (mmol/L)	1.04–1.89	Low HDL-C Men < 1.03, Women < 1.29	77 (33.3)	72 (37.3)	5 (13.2)
		Normal	154 (66.7)	121 (62.7)	33 (86.8)
Serum magnesium (mmol/L)	0.74–0.99	Low Status < 0.74	77 (33.3)	62 (32.1)	15 (39.5)
		Normal	154 (66.7)	131 (67.9)	23 (60.5)

Abbreviations: SBP, systolic blood pressure; DBP, diastolic blood pressure; FBG, fasting blood glucose; TG, triglyceride; HDL-C, high-density lipoprotein cholesterol; LDL-C, low-density lipoprotein cholesterol; WC, waist circumference. * The cutoff points are based on the International Diabetes Federation criteria for metabolic syndrome diagnosis. ** The normal reference ranges are based on those used by the Kuwaiti Ministry of Health.

**Table 4 nutrients-14-05257-t004:** Cardiometabolic factors among study participants.

Cardiometabolic Risk Factors	Overall Sample, *n* = 231	Without MetS, *n* = 180	With MetS *n* = 51	Sig ^a^
	Median [Interquartile Range]			
Age, years	28.00 [22.00–35.00]	27.00 [22.00–33.00]	35.00 [29.00–47.00]	0.000
BMI, kg/m^2^	24.90 [22.50–27.68]	24.06 [22.15–25.71]	28.12 [26.34–31.14]	0.000
WC, cm	85.00 [78.00–90.00]	81.00 [78.00–88.00]	89.50 [86.00–96.00]	0.000
SBP, mmHg	120.00 [118.00–123.00]	120.00 [117.25–122.00]	122.00 [119.00–134.00]	0.000
DBP, mmHg	80.00 [78.00–82.00]	79.50 [77.00–81.00]	82.00 [79.00–85.00]	0.000
FBG, mmol/L	5.00 [4.62–5.51]	4.85 [4.56–5.15]	5.80 [5.50–6.49]	0.000
TG, mmol/L	0.99 [0.71–1.51]	0.87 [0.64–1.19]	1.84 [1.26–2.60]	0.000
HDL-C, mmol/L	1.35 [1.12–1.56]	1.42 [1.22–1.60]	1.03 [0.89–1.24]	0.000
LDL-C, mmol/L	2.49 [2.01–2.94]	2.40 [1.94–2.85]	2.77 [2.17–3.31]	0.008
Total cholesterol, mmol/L	4.43 [3.78–5.12]	4.28 [3.72–5.05]	4.76 [4.22–5.62]	0.003
Serum magnesium level, mmol/L	0.79 [0.73–0.86]	0.81 [0.76–0.90]	0.70 [0.63–0.74]	0.000

^a^ Significance level using independent-sample Mann–Whitney U-test, *p* < 0.05.

**Table 5 nutrients-14-05257-t005:** Spearman’s correlations with serum magnesium level as the dependent variable.

	Correlation Coefficients, r-Value
BMI, kg/m^2^	−0.423 **
WC, cm	−0.412 **
SBP, mmHg	−0.194 **
DBP, mmHg	−0.188 **
FBG, mmol/L	−0.517 **
TG, mmol/L	−0.586 **
HDL-C, mmol/L	0.368 **
LDL-C, mmol/L	−0.265 **
Total cholesterol, mmol/L	−0.280 **

All variables entered in continuous scale, for total population *n* = 231; ** *p* < 0.001.

**Table 6 nutrients-14-05257-t006:** Binary logistic regression analysis with hypomagnesemia as the independent variable.

	Model Type	Sig	Serum Magnesium Categories 1 = Serum Magnesium < 0.74 mmol/L			
			Exp(B)	95% CI		Nagelkerke R Square
				Lower	Upper	
IDF Criteria						
Overall sample *n* = 51/231	Unadjusted	0.000	16.40	7.50	35.84	0.334
	Adjusted ^a^	0.000	11.70	4.87	28.14	0.364
Men only *n* = 6/38	Unadjusted	0.039	11.00	1.13	106.84	0.192
	Adjusted ^b^	0.575	--	--	--	--
Women only *n* = 45/193	Unadjusted	0.000	18.77	8.07	43.66	0.192
	Adjusted ^c^	0.000	13.36	5.27	33.83	0.385
ATP III criteria						
Overall sample *n* = 35/231	Unadjusted	0.000	9.86	4.21	23.10	0.189
	Adjusted ^a^	0.001	5.44	2.10	14.10	0.260
Men only *n* = 3/38	Unadjusted	0.999	--	--	--	--
	Adjusted ^b^	0.999	--	--	--	--
Women only *n*= 32/193	Unadjusted	0.000	9.71	4.03	23.39	0.204
	Adjusted ^c^	0.001	5.27	1.99	13.99	0.252

Dependent variable with (coded 1) and without (coded 0) metabolic syndrome. ^a^ age, sex, and BMI; ^b^ age and BMI only; ^c^ age, BMI, and any birth control or having regular period.

**Table 7 nutrients-14-05257-t007:** Logistic regression (both adjusted and unadjusted) for MetS components (dependent variables) with hypomagnesemia (independent variable).

Dependent Variable ^a^	Model Type	Serum Magnesium Categories, 1 = Serum Magnesium Level < 0.74 mmol/L				
		Sig	Exp(B)	95% CI		Nagelkerke R Square
				Lower	Upper	
IDF Obese waist	Unadjusted	0.000	5.08	2.96	9.61	0.163
IDF Obese waist	Adjusted ^b^	0.005	3.62	1.48	8.85	0.236
ATP III Obese waist	Unadjusted	0.001	2.93	1.54	5.54	0.064
ATPIII Obese waist	Adjusted ^b^	0.572	--	--	--	--
Elevated TG level	Unadjusted	0.000	12.74	5.67	28.63	0.259
Elevated TG level	Adjusted ^b^	0.000	9.29	3.97	21.73	0.342
Elevated FBG	Unadjusted	0.000	10.57	5.13	21.77	0.261
Elevated FBG	Adjusted ^b^	0.000	7.01	3.25	15.11	0.323
Elevated Systolic BP	Unadjusted	0.000	5.69	2.67	12.14	0.126
Elevated Systolic BP	Adjusted ^b^	0.022	2.88	1.16	7.11	0.221
Elevated Diastolic BP	Unadjusted	0.000	7.53	2.85	19.89	0.113
Elevated Diastolic BP	Adjusted ^b^	0.021	3.64	1.22	10.89	0.223
Low HDL level	Unadjusted	0.000	3.13	1.76	5.58	0.089
Low HDL	Adjusted ^b^	0.000	3.27	1.70	6.32	0.259

^a^ Cardiometabolic high-risk category coded 1, whereas normal category = 0. ^b^ Adjusted for age, sex, and BMI.

## Data Availability

Upon reasonable request from the corresponding author.

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
