# Peer review of "Hypomagnesemia and the Metabolic Syndrome among Apparently Healthy Kuwaiti Adults: A Cross-Sectional Study"

_nutrients, 2022, doi:10.3390/nu14245257_

Round 1

Reviewer 1 Report (Previous Reviewer 2)

Dear authors:

In your article, you investigated the prevalence of hypomagnesemia, metabolic syndrome and its components in 231 healthy adults, as well as the relationship between hypomagnesemia and metabolic syndrome. The study in very interesting and you have valuable data, but I think that you have some flaws that need to be improved before your article can be published. First, I think that your sampling method is inadequate to estimate prevalences, so I believe that you should focus on the relationship between hypomagnesemia and metabolic syndrome and its components, and not on the prevalence of MetS or hypomagnesemia. Second, I think that with your experimental design it is not possible to establish the odd of developing metabolic syndrome or its components based on magnesium status, because magnesium status is concurrent with the presence of metabolic syndrome. In this regard, you can only estimate the odd that people with hypomagnesemia have MetS or viceversa.

Considering the previously mentioned, some specific commentaries are:

- In the abstract, the phrase "hypomagnesemia is a risk factor for MetS and its individual components among adults in Kuwait" can not be concluded with a cross sectional design. Hypomagnesemia could be a consequence of the pathophysiology of MetS or some of its components and not be a risk factor.

- Due to the characteristics of the sampling method, it is not possible to establish the prevalence of MetS in healthy people in Kuwait. Recruitment was performed in hospitals and the sample was selected by convenience of the researchers.

- In the subsection 3.6 "Correlations of hypomagnesemia with MetS and its components", please change all the terms"odds of developing" for "odds of having".

- In the Discussion section, it is incorrect to affirm that "hypomagnesemia increased the odds of developing these risk factors, independently from age, sex, and BMI" (lines 300-301) due to the previously mentioned. Please change for odds of having.

- In table 2, check relative values of Irregular Menstrual Period.

- In table 3, check relative values of TG level and LDL-cholesterol in females and LDL-cholesterol in males.

- References 19 and 24 are the same.

Author Response

Reviewer 2 Report (New Reviewer)

1.       This is an interesting study on the implication of magnesium in the genesis of metabolic syndrome and cardiovascular diseases. However, cross-sectional studies collect data at a single point in time, and they function as "snapshots" of the study population at the specified time, making it difficult to conclude over the implication of magnesium in pathogenic mechanisms that produce progressively, during time (alterations of lipid and glucose metabolism for example). The conclusion is that cross-sectional studies are not the most appropriate way to test a hypothesis, and the authors should explain the choice and how the study's rationale is compatible with the cross-sectional analysis of data.

2.       Is the unidirectional relationship of hypomagnesemia as a risk factor for metabolic syndrome appropriate? How is the presumptive inverse relationship of hypomagnesemia being induced by various modifications or comorbidities associated with metabolic syndrome corrected in your study?

3.       Why do the authors assume and how can this study's data prove that the findings reflecting the association between hypomagnesemia and cardiometabolic factor respect a causal relationship?

4.       Evidence-based magnesium-containing supplements recommendations should be included in the text.

5.       Rows 240-242 authors should consider changing the wording- "Serum magnesium level showed a significant negative association with all the cardiometabolic risk factors (p<0.001) (Table 5), except for HDL-C level, which showed a positive association (r=0.368, p<0.001)" as it appears from the text that  HDLc is also a cardiometabolic risk factor.

6.       Row 248- the term "risk" should be reconsidered as the risk implies a causal relationship between hypomagnesemia which is not demonstrated until this point in the study. This idea keeps repeating throughout the text and should be better documented or redefined. On the other hand, the association between hypomagnesemia and the occurrence of MetS and its components clearly emerges from the data described.

Round 2

Reviewer 2 Report (New Reviewer)

1.       AUTHORS’ RESPONSE: We would like to thank the reviewer for his valuable insights and comments which helped improve the manuscript immensely. All these points have been considered in the revision, and the limitation of the cross-sectional design is provided in Line 423-424.

REVIEWER COMMENT: This point was appropriately accomplished.

2.       AUTHORS’ RESPONSE: Hypomagnesemia was the independent variable and METS and its components individually were the dependent variable in different models. This is an appropriate assumption based on previous studies and knowledge of mechanisms. By selection criteria we excluded subjects with any current disease or taking meds that may interfere with Mg levels. If there were residual confounding that we did not account for the in the adjusted models, our estimates should be affected negatively.

REVIEWER COMMENT: I strongly recommend referring shortly (somewhere in the first part of the review article) to the specific components of the metabolic syndrome identified to date to be affected by magnesium deficiency. It would be helpful for a clearer perspective.

3.       Evidence-based magnesium-containing supplements recommendations should be included in the text. This was briefly mentioned in the conclusions line 442

REVIEWER COMMENT: Point appropriately accomplished.

4.       Evidence-based magnesium-containing supplements recommendations should be included in

the text. This was briefly mentioned in the conclusions line 442

REVIEWER COMMENT: Done.

5.       This sentence is correct and expected because HDL-C only has a positive relationship with serum magnesium, and when both are increased, they decrease CVD risk.

REVIEWER COMMENT: Clarification provided. Accomplished point.

6.       The wording has been revised to remove any definitive implication of risk.

REVIEWER COMMENT: DONE.

Author Response

REVIEWER COMMENT: I strongly recommend referring shortly (somewhere in the first part of the review article) to the specific components of the metabolic syndrome identified to date to be affected by magnesium deficiency. It would be helpful for a clearer perspective.

RESPONSE:

Dear Reviewer,

Thank you for your valuable suggestion; the following sentence was added in the introduction Lines 43-44: "Hypomagnesemia has been associated with the components of MetS, including hyperglycemia, hypertension, hypertriglyceridemia and insulin resistance [5]". This was in addition to the preexisting following sentence in lines 45-47: "Magnesium supplementation reportedly demonstrates good efficacy on the components of MetS, including insulin sensitivity, fasting blood glucose (FBG), triglyceride (TG) levels, high-density lipoprotein cholesterol (HDL-C) levels, and high blood pressure (BP) [6]. "

This manuscript is a resubmission of an earlier submission. The following is a list of the peer review reports and author responses from that submission.

Round 1

Reviewer 1 Report

Well-written and easy to read manuscript!

This paper further adds to the growing literature regarding electrolyte deficiencies and cardiometabolic health risks around the world. 

In brief, this paper found significant associations between low Mg2+ and higher cardiometabolic risk factors (MetS).

Here are the following recommendations:

- Since Mg2+ is a part of the regular CMP panel, would like to understand the rationale why the investigators chose to investigate Mg2+ alone?

- Would recommend that other electrolytes (Na+, Cl-, K+) be reported in Table 2 & 3.

- Since a significant source for Mg2+ is food intake, it is important to provide details regarding subjects' diets, at the minimal average caloric intake daily. This is also extremely important since diet is significant associated with MetS. Without adjusting for average caloric intake, statistical analyses may be skewed/overestimated.

- Since many variables are significantly different between MetS vs non-MetS (by criteria definition), it is hard to say whether Mg2+ is truly correlated or that it is correlated with an unexplored factor that has significant associations with MetS. Would recommend running the analyses with all major electrolytes or adjusting for some of them as well as average caloric intake.

Author Response

Comments and Suggestions for Authors

Authors Responses

Reviewer #1

Well-written and easy to read manuscript!

Thank you

This paper further adds to the growing literature regarding electrolyte deficiencies and cardiometabolic health risks around the world. In brief, this paper found significant associations between low Mg2+ and higher cardiometabolic risk factors (MetS).

This article is specifically targeting serum magnesium and its deficiency in relation to cardiometabolic health, and not including other electrolytes. This is in effort to understand hypomagnesemia in relation to MetS and to establish the groundwork for an interventional study and dietary analysis- using serum magnesium to assess status.

- Since Mg2+ is a part of the regular CMP panel, would like to understand the rationale why the investigators chose to investigate Mg2+ alone?

Serum Mg2+ is a valid biomarker for magnesium status and has been used in several investigations in association with metabolic disorders (Pelczy´nska et al., 2022) [Ref 7]. A low magnesium status, which is often underdiagnosed, potentiates the reactivity to various immune challenges and is implicated in the pathophysiology of many common chronic diseases (Maier et al., 2021) [Ref 8]. We wanted to investigate hypomagnesemia as a risk factor for MetS and its components in apparently healthy adults. This work aligns with the current interests in measuring magnesium status and its relationship to cardiometabolic risk factors and glucose homeostasis as a basis for interventional studies [Lines 34-43]. Even if the regular CMP panel contains all electrolytes, we obtained specific permission from the Ministry of Health for the specific data needed to support this study's hypotheses, and serum Mg2+ was needed only. The interaction between Mg2+ and other electrolytes K+, P+, Ca2+ require further investigations.

- Would recommend that other electrolytes (Na+, Cl-, K+) be reported in Table 2 & 3.

This data was not obtained and is not part of the current investigation. The interaction of these electrolytes with Mg2+ and cardiometabolic health would require more detailed work, not in the scope of the current investigation.

- Since a significant source for Mg2+ is food intake, it is important to provide details regarding subjects' diets, at the minimal average caloric intake daily. This is also extremely important since diet is significant associated with MetS. Without adjusting for average caloric intake, statistical analyses may be skewed/overestimated.

We agree with the reviewer that total energy intake can be important to measure when adjusting for dietary Mg2+. However, this data is not available because, with best efforts, the reported average caloric intake is flawed by underreporting, which can significantly affect estimates, as well (Poslusna et al., 2009). Given the focus in this study was serum Mg2+ status, in our analyses, BMI, which is a predictor of energy intake (Trijsburg et al., 2017) was used for the adjusted models. Our estimates were improved or decreased slightly and not attenuated. The estimates we obtained were within the range of previous reports and somewhat lower because of early detection in apparently healthy subjects free of chronic disease [Lines 280-287].  

Per your suggestions, I added the following sentences to this study's limitations: “Hypokalemia and hypocalcemia are seen with poor magnesium status, and so the lack of the above measures could be a limitation. Future work could more comprehensively assess Mg2+ status,  oral magnesium load test, food records of Mg intake, and urinary magnesium measures for a more comprehensive assessment of Mg status”. [Lines 377-381]

-  Poslusna, K., Ruprich, J., De Vries, J., Jakubikova, M., & Van't Veer, P. (2009). Misreporting of energy and micronutrient intake estimated by food records and 24 hour recalls, control and adjustment methods in practice. British Journal of Nutrition, 101(S2), S73-S85.

-  Trijsburg L, Geelen A, Hollman PC, Hulshof PJ, Feskens EJ, Van't Veer P, et al. BMI was a consistent determinant related to misreporting of energy, protein and potassium intake using self-report and duplicate portion methods. Public Health Nutr. 2017;20(4):598-607.

-  Rupert W Jakes, Nicholas E Day, Robert Luben, Ailsa Welch, Sheila Bingham, Jo Mitchell, Susie Hennings, Kirsten Rennie, Nicholas J Wareham. Adjusting for energy intake—what measure to use in nutritional epidemiological studies? International Journal of Epidemiology, Volume 33, Issue 6, December 2004, Pages 1382–1386.

- Since many variables are significantly different between MetS vs non-MetS (by criteria definition), it is hard to say whether Mg2+ is truly correlated or that it is correlated with an unexplored factor that has significant associations with MetS. Would recommend running the analyses with all major electrolytes or adjusting for some of them as well as average caloric intake.

Adding more variables changes what we intended to do.

Based on the physiological mechanism for Mg2+ and the previous literature, the study design was constructed. The research question was whether hypomagnesemia was associated with MetS and its components. We cannot eliminate residual confounding. This is also the case with even bigger cohorts and prospective designs, with more variables measured and used in their analyses, including dietary data. Studies were reporting similar findings as our study, and others reported no associations because of variabilities between populations.  We have addressed our research question thoroughly to the best of our knowledge.

-     Guerrero-Romero F, Rodríguez-Morán M. Low serum magnesium levels and metabolic syndrome. Acta Diabetol. 2002 Dec;39(4):209-13.

doi: 10.1007/s005920200036.

-     Gohari-Kahou, M., Darroudi, S., Saberi-Karimian, M., Parizadeh, S.-M., Asadi, Z., Javandoost, A., Safarian, M., Mouhebati, M., Ebrahimi, M., Ferns, G. A., Kazerani, H. R., & Ghayour-Mobarhan, M. (2020). The association between serum and dietary magnesium with cardiovascular disease risk factors in Iranian adults with metabolic syndrome. Translational Metabolic Syndrome Research, 3, 42–48. https://doi.org/10.1016/J.TMSR.2020.10.001

Reviewer 2 Report

Dear authors,

In this article, you propose to evaluate the prevalence of metabolic syndrome in an apparently healthy population in Kuwait and its relationship with hypomagnesemia, which is not too original but adds to a topic that has been explored. Although the approach to the subject is complex due to the large number of factors that cannot be controlled, I believe that the main problems of your study are the exclusion criteria and the characteristics of the sampling, which prevents you from determining the prevalences that you propose.  For this reason, I suggest restricting the objective of the study to evaluating the relationship between hypomagnesemia and metabolic syndrome only. Some specific comments are:

Introduction:

- Lines 43 and 44 indicate that magnesium depletion can induce inflammation directly and indirectly by modifying the microbiota. However, it is not the only mechanism proposed to explain this relationship (Seminars in Cell & Developmental  Biology, 2021; 115: 37-44) . Please complete.

- On line 61, delete Al Zenki et al., 2012.

- Line 62 indicates that Sorkhou et al reported a prevalence of metabolic syndrome of 34%, but that in men it was 43.5% and in women it was 56.4%, both higher than the general prevalence. Check please.

- Line 67 indicates that you aimed to address the prevalence of hypomagnesaemia in a sample of adults living in Kuwait. However, you did not determine the sample size necessary for the calculation to have statistical value, nor did you use an appropriate type of sampling.

- On line 71, you indicate that your specific objective was to determine the prevalence of metabolic syndrome in a sample of healthy individuals. Again, there is no sample size calculation or adequate sampling type to determine prevalence.

Materials and Methods:

- The reported prevalences are possibly underestimated due to the exclusion criteria, since people with diabetes or treatment for hypertension or dyslipidemia are possibly part of the group with the highest prevalence of metabolic syndrome (lines 93 to 98). In addition, due to the exclusion criteria, the relationship between metabolic syndrome and hypomagnesemia can only be determined in a very limited group of patients. In this case, only individuals not diagnosed with fasting hyperglycemia or diabetes, hypertension or dyslipidemia are included, which may be in early stages of the diseases or with very poor medical control.

- Regarding the statistical analysis, why you didn't use a stepwise regression analysis instead of first testing the factors in univariate analyses?

Results:

- Line 172 indicates that postmenopausal women had a higher prevalence of metabolic syndrome than women of childbearing age. First, the number of postmenopausal women is very small. Second, how did you correct that result based on age?

- On line 231, I suggest replacing developing with having.

- In Table 1, under men in Annual Family Income, the numbers don't fit me.

- In Table 2, in women without metabolic syndrome and with irregular menstrual periods, the absolute number and prevalence are reversed.

- In Table 3, in women with high LDL cholesterol, the prevalence is to 2 decimal places and all the others are to only one.

Conclussion:

- In the conclusions, I would take out the phrase sex- and age-specific waist circumference cut-off points are required since the subject was not addressed before.

Author Response

Comments and Suggestions for Authors

Authors Responses

Review #2

In this article, you propose to evaluate the prevalence of metabolic syndrome in an apparently healthy population in Kuwait and its relationship with hypomagnesemia, which is not too original but adds to a topic that has been explored. Although the approach to the subject is complex due to the large number of factors that cannot be controlled, I believe that the main problems of your study are the exclusion criteria and the characteristics of the sampling, which prevents you from determining the prevalences that you propose.  For this reason, I suggest restricting the objective of the study to evaluating the relationship between hypomagnesemia and metabolic syndrome only.

We thank the reviewer for this suggestion, and we would like to clarify that choosing the exclusion criteria intentionally recruited participants who are apparently healthy and not yet prescribed medications that affect their serum Mg2+ levels. Thus, it is important to evaluate the associations between hypomagnesemia and the individual components of MetS at an early stage before the clustering of many risk factors and comorbid inflammatory conditions such as diabetes and kidney dysfunction.  Early detection of the relationship between serum Mg2+ and MetS and its components underscores the early detection of hypomagnesemia as a risk factor for metabolic dysregulation. (Pelczy´nska et al., 2022) [Ref 7].

Some specific comments are:

Introduction: Lines 43 and 44 indicate that magnesium depletion can induce inflammation directly and indirectly by modifying the microbiota. However, it is not the only mechanism proposed to explain this relationship (Seminars in Cell & Developmental  Biology, 2021; 115: 37-44) . Please complete.

(Added this sentence in the introduction, Lines 45-48): “In addition, magnesium deficiency promotes inflammation by dysregulating the immunological function through priming phagocytes, enhancing granulocyte oxidative burst, activating endothelial cells, and increasing the levels of cytokines production”. Also, the suggested reference is added [Ref 8].

On line 61, delete Al Zenki et al., 2012.

Done [Ref 12].

- Line 62 indicates that Sorkhou et al reported a prevalence of metabolic syndrome of 34%, but that in men it was 43.5% and in women it was 56.4%, both higher than the general prevalence. Check please.

Thank you for pointing this mistake out. It was corrected as follows: [Line 65: in men, it was 28.7%, and in women, it was 39.7%].

- Line 67 indicates that you aimed to address the prevalence of hypomagnesaemia in a sample of adults living in Kuwait. However, you did not determine the sample size necessary for the calculation to have statistical value, nor did you use an appropriate type of sampling.

- On line 71, you indicate that your specific objective was to determine the prevalence of metabolic syndrome in a sample of healthy individuals. Again, there is no sample size calculation or adequate sampling type to determine prevalence.

We employed a convenient sampling method [Line 79], due to difficulties with recruitment and timing of the study, as permitted by the participating hospitals. We continued recruitment from May 2018 to January 2019, for the duration of the ethical approval. We obtained significant results even with not reaching sample size target. The sample size was priori calculated based on previously reported % hypomagnesemia and MetS in adults. The Sample size calculated ranged 240-370 patients using the single proportion equation:

 With a standard normal variate at 5% type 1 error (P<0.05), Z1- α/2 = 1.96. The range of p value= 0.2 - 0.4 based on a previous study which is expected proportion in population.  d value of 0.05 which is the absolute error or precision.

Materials and Methods:

- The reported prevalences are possibly underestimated due to the exclusion criteria, since people with diabetes or treatment for hypertension or dyslipidemia are possibly part of the group with the highest prevalence of metabolic syndrome (lines 93 to 98). In addition, due to the exclusion criteria, the relationship between metabolic syndrome and hypomagnesemia can only be determined in a very limited group of patients. In this case, only individuals not diagnosed with fasting hyperglycemia or diabetes, hypertension or dyslipidemia are included, which may be in early stages of the diseases or with very poor medical control.

We intentionally targeted individuals with no overt diseases and taking medications affecting Mg2+ levels to estimate the prevalence in apparently healthy individuals, which is important to establish relationship between Mg2+ and MetS and its components even before the clustering of metabolic dysregulations.  Our data was consistent with Badr et al [ref 20] as discussed this in Lines 281-282. Investigators also used “apparently healthy” sample. The possibility of selection bias mentioned in the limitation [Line 377-379].

- Regarding the statistical analysis, why you didn't use a stepwise regression analysis instead of first testing the factors in univariate analyses?

Because both the dependent variable (with and without MetS) (or with and without any cardiometabolic components) and the independent variable (with/without hypomagnesemia) were binary outcomes. So, we used univariate logistic regression – to establish the associations- and then multivariate logistic regression to adjust for covariates age, sex, and BMI. The covariates were chosen according to the Chi-Square analyses.  Stepwise Linear regression would be used for continuous variables.

Results: - Line 172 indicates that postmenopausal women had a higher prevalence of metabolic syndrome than women of childbearing age. First, the number of postmenopausal women is very small. Second, how did you correct that result based on age?

[Line 177-180] Among women who reported hormonal irregularities, those with menopause had higher prevalence of the metabolic syndrome. They would be a small percent of the group. These comparisons were not adjusted for age, as this would be overadjustment. These hormonal dysregulations become more common with age. Adjustment for age were done in the multivariate logistic regression models in Table 6 & 7.

- On line 231, I suggest replacing developing with having.

Done

- In Table 1, under men in Annual Family Income, the numbers don't fit me.

Done

{Thanks for the correction}

- In Table 2, in women without metabolic syndrome and with irregular menstrual periods, the absolute number and prevalence are reversed.

Done

{Thanks for the correction}

- In Table 3, in women with high LDL cholesterol, the prevalence is to 2 decimal places and all the others are to only one.

Done

{Thanks for the correction}

Conclussion:

- In the conclusions, I would take out the phrase sex- and age-specific waist circumference cut-off points are required since the subject was not addressed before.

This point was discussed in Lines 273-279, that a discrepancy in MetS prevalence based on the two definitions used is consistent with that observed in other investigations, with IDF-defined MetS showing a higher prevalence than ATP III-defined MetS. due to the lower cutoff point values for waist circumference used in the IDF. For example, the waist circumference in females is ≥ 80 cm and > 88 cm based on IDF and ATP III criteria, respectively. Therefore, this conclusion is warranted.

Round 2

Reviewer 2 Report

Dear authors:

Thank you very much for correcting many of the suggestions made. However, I would like to state again some points that I consider relevant:

- I understand the purpose of relating hypomagnesemia to metabolic syndrome free of interference, including drugs. That is not the problem. However, I maintain the point that the experimental design is not adequate to determine the prevalence of hypomagnesemia or metabolic syndrome, even in "apparently healthy" individuals, due to the way the sample was selected. The sample is for convenience, you did not reach the sample size calculated previously in the sample size calculation (even if you obtained statistical significance regarding the association between hypomagnesemia and metabolic syndrome), and is not representative unless you have included criteria of representativeness that have not been declared in the methodology. For this reason, prevalences cannot be estimated.

-The discussion is mainly focused in prevalences and although you stated that you can not assume causality, all discussion is directed toward the effect of hypomagnesemia on metabolic syndrome.

- Regarding to the conclusion, I would like to keep the point too. There are cut-off points by sex, and even you mentioned them. You did not justify why you require specific cut-off values for Kuwaiti people. Regarding the cut-off points for age, I don't think the point is adequately justified in the text either. The cut-off points relate the parameter to the risk of develop some condition, and you did not explain how age change the sensitivity of the cut-off points with respect to the risk of cardiovascular disease, for example.

Author Response

Dear authors:

Thank you very much for correcting many of the suggestions made. However, I would like to state again some points that I consider relevant:

I thank the reviewer for his attentiveness and thorough review, which contributed to improving the quality and presentation of this article.

- I understand the purpose of relating hypomagnesemia to metabolic syndrome free of interference, including drugs. That is not the problem.

However, I maintain the point that the experimental design is not adequate to determine the prevalence of hypomagnesemia or metabolic syndrome, even in "apparently healthy" individuals, due to the way the sample was selected. The sample is for convenience, you did not reach the sample size calculated previously in the sample size calculation (even if you obtained statistical significance regarding the association between hypomagnesemia and metabolic syndrome), and it is not representative unless you have included criteria of representativeness that have not been declared in the methodology. For this reason, prevalences cannot be estimated.

I understand the reviewer's concern. The key distinction is the two potential populations: the study sample and the target population. The study sample is the people you see, and the target population is the overall set of people you want to draw inferences about. Obtaining a convenience sample gives the prevalence for the study sample, but the study sample prevalence may not be the same as the target population. This study sample prevalence is not generalizable to the target population prevalence. Both are still prevalences, the difference is the population to which it refers. We clearly stated this in the study limitations [Lines 404-407]. Our study has a few limitations. The cross-sectional study design cannot infer causality. In addition, the study included a small sample size that covered only the Al-Ahmadi and Hawalli governorates. Thus, the sample is not representative of the entire Kuwait population.

We added to the methods section [Line 84-87] The sample size was priori calculated based on previously reported % hypomagnesemia and MetS in adults and ranged from 240-370 participants.

We added to the discussion the following [Line 407-409]: This study sample size (n=231) fell below the lower range of the priori calculated target (240 participants) due to difficulties with recruitment, which further limits the generalizability of the findings.

-The discussion is mainly focused in prevalences and although you stated that you can not assume causality, all discussion is directed toward the effect of hypomagnesemia on metabolic syndrome.

Based on the variables investigated and the results, the discussion was constructed. [Line 258-271] Hypomagnesemia, Mg2+ body pools, and related health conditions. [Line 273-297] MetS definitions and comparisons to previous reports in GCC; and variations caused by sampling criteria. [Line 299-308] Weight and age covariates for `MetS and CVD. [Line 310-328] Higher prevalence in women and covariates: adiposity, hormonal dysregulations, and Mg2+ serum levels. [Line 330-342] Menopause/pregnancy life stages affecting adiposity and risks of MetS. [Line 344-367] Hypomagnesaemia in relation to all components of MetS, and in relation to inflammation and metabolic dysregulations. [Line 369 -383] Strengths and limitations. [Line 384-399] Conclusions.

- Regarding to the conclusion, I would like to keep the point too. There are cut-off points by sex, and even you mentioned them.

You did not justify why you require specific cut-off values for Kuwaiti people.

In the discussion where we mentioned the discrepancy between estimation of %MetS using both definitions [Line 277-282] In this study, we found that the prevalences of IDF- and ATP III-defined MetS were 22% and 15% in the overall sample population, respectively. The discrepancy in MetS prevalence based on the two definitions used is consistent with that observed in other investigations, with IDF-defined MetS showing a higher prevalence than ATP III-defined MetS. This may be due to the lower cutoff point values for waist circumference used in the IDF. For example, the waist circumference in females is ≥ 80 cm and > 88 cm based on IDF and ATP III criteria, respectively.

We added these points to the discussion [Line283-288] Until more specific data are available from this region, the European WC cutoff points are used for Middle Eastern and Arab countries [21]. Studies in Iran, Oman, Iraq, Tunisia, and Egypt proposed various population-specific cutoff point values reflective of differences in fat distribution patterns according to sex and between ethnicities, as suggested by IDF [22]. Such endeavor may improve the estimation of abdominal obesity and MetS prevalence among Kuwaitis.

21 Alberti KG, Zimmet P, Shaw J. Metabolic syndrome--a new world-wide definition. A Consensus Statement from the International Diabetes Federation. Diabet Med. 2006 May;23(5):469-80. doi: 10.1111/j.1464-5491.2006.01858.x.

22 Assaad-Khalil SH, Mikhail MM, Aati TA, Zaki A, Helmy MA, Megallaa MH, Hassanien R, Rohoma KH. Optimal waist circumference cutoff points for the determination of abdominal obesity and detection of cardiovascular risk factors among adult Egyptian population. Indian J Endocrinol Metab. 2015 Nov-Dec;19(6):804-10. doi: 10.4103/2230-8210.167556

Regarding the cut-off points for age, I don't think the point is adequately justified in the text either. The cut-off points relate the parameter to the risk of develop some condition, and you did not explain how age change the sensitivity of the cut-off points with respect to the risk of cardiovascular disease, for example.

In Table 2, we provided % MetS according to age groups in the study sample, and there was a statistically significant increase from the youngest group, 18-29 yrs age group = 11.7%, middle group, 30-49 yrs group =26.6%, and to oldest group 50-65= 92.3% [Lines 171-174].

We now added age-adjusted % MetS (Table-2), which has improved the sensitivity of diagnosing MetS. [Lines175-178] The age-adjusted prevalence of MetS according to IDF for the overall sample increased from 22.1% to 27.4% (from 23.3% to 28.3% in females); and (from 15.8% to 23% in males). Similarly, when using ATP III, the overall MetS prevalence increased from 15.2% to 19.1% (from 16.6% to 20.6% in females); and (from 7.9% to 11.9% in males).

We added the following paragraph to the discussion to explain the relationship between age and risk of CVD mediated through MetS [Lines346-365] Overeating and sedentary lifestyle worsens with age and mediates MetS by increasing the accumulation of central adiposity and ectopic fat infiltration in the skeletal muscle and the liver, driving the development of insulin resistance with ectopic fat accumulation, magnesium metabolism alterations, systemic and hypothalamic inflammation, shortening of telomeres length, dysregulating epigenetics mechanisms, and disturbing circadian rhythm [38]. Several population-based studies showed that older adults (older age group had a higher prevalence of MetS than younger age groups, and that the MetS factors were equally prevalent in men and women [38-39]. It was suggested that the diminished sex differences in the metabolic risk profile might be due to the diminished sex differences in total and visceral adiposity with age and the cardiometabolic effects of menopause. However, there appeared to be important sex and age differences in the way the different metabolic syndrome combinations relate to CVD morbidity and mortality risk [39]. For example, although HDL was one of the most common metabolic syndrome components in younger women and among the least common in older women, it was one of the more robust correlates of mortality risk in both age strata [39].  Moreover, the association between MetS and mortality risk did not appear to be related to the number of MetS components one displays [39-40]. More research with comprehensive data is needed to determine the specific combinations of metabolic syndrome components predictive of CVD morbidity and mortality risk in Kuwait.

38- Dominguez, Ligia J.; Barbagallo, Mario. The biology of the metabolic syndrome and aging. Current Opinion in Clinical Nutrition and Metabolic Care: January 2016 - Volume 19 - Issue 1 - p 5-11 doi: 10.1097/MCO.0000000000000243

39 Kuk JL, Ardern CI. Age and sex differences in the clustering of metabolic syndrome factors: association with mortality risk. Diabetes Care. 2010 Nov;33(11):2457-61. doi: 10.2337/dc10-0942.

40 Ge, H., Yang, Z., Li, X. et al. The prevalence and associated factors of metabolic syndrome in Chinese aging population. Sci Rep 10, 20034 (2020). https://doi.org/10.1038/s41598-020-77184-x
